# Quality and Process Optimization of Infrared Combined Hot Air Drying of Yam Slices Based on BP Neural Network and Gray Wolf Algorithm

**DOI:** 10.3390/foods13030434

**Published:** 2024-01-29

**Authors:** Jikai Zhang, Xia Zheng, Hongwei Xiao, Chunhui Shan, Yican Li, Taoqing Yang

**Affiliations:** 1College of Mechanical and Electrical Engineering, Shihezi University, Shihezi 832003, China; 15154159806@163.com (J.Z.); 20202109079@stu.shzu.edu.cn (Y.L.); 15299911236@163.com (T.Y.); 2Key Laboratory of Northwest Agricultural Equipment, Ministry of Agriculture and Rural Affairs, Shihezi 832003, China; 3Key Laboratory of Modern Agricultural Machinery Corps, Shihezi 832003, China; 4College of Engineering, China Agricultural University, Beijing 100080, China; xhwcaugxy@163.com; 5College of Food, Shihezi University, Shihezi 832003, China; sch_0909@163.com

**Keywords:** yam slices, infrared combined hot air drying, drying characteristics, BP neural network coupled with gray wolf algorithm, multi-objective optimization

## Abstract

In this paper, the effects on drying time (Y_1_), the color difference (Y_2_), unit energy consumption (Y_3_), polysaccharide content (Y_4_), rehydration ratio (Y_5_), and allantoin content (Y_6_) of yam slices were investigated under different drying temperatures (50–70 °C), slice thicknesses (2–10 mm), and radiation distances (80–160 mm). The optimal drying conditions were determined by applying the BP neural network wolf algorithm (GWO) model based on response surface methodology (RMS). All the above indices were significantly affected by drying conditions (*p* < 0.05). The drying rate and effective water diffusion coefficient of yam slices accelerated with increasing temperature and decreasing slice thickness and radiation distance. The selection of lower temperature and slice thickness helped reduce the energy consumption and color difference. The polysaccharide content increased and then decreased with drying temperature, slice thickness, and radiation distance, and it was highest at 60 °C, 6 mm, and 120 mm. At 60 °C, lower slice thickness and radiation distance favored the retention of allantoin content. Under the given constraints (minimization of drying time, unit energy consumption, color difference, and maximization of rehydration ratio, polysaccharide content, and allantoin content), BP-GWO was found to have higher coefficients of determination (*R*^2^ = 0.9919 to 0.9983) and lower *RMSEs* (reduced by 61.34% to 80.03%) than RMS. Multi-objective optimization of BP-GWO was carried out to obtain the optimal drying conditions, as follows: temperature 63.57 °C, slice thickness 4.27 mm, radiation distance 91.39 mm, corresponding to the optimal indices, as follows: Y_1_ = 133.71 min, Y_2_ = 7.26, Y_3_ = 8.54 kJ·h·kg^−1^, Y_4_ = 20.73 mg/g, Y_5_ = 2.84 kg/kg, and Y_6_ = 3.69 μg/g. In the experimental verification of the prediction results, the relative error between the actual and predicted values was less than 5%, proving the model’s reliability for other materials in the drying technology process research to provide a reference.

## 1. Introduction

Yam is a food rich in biologically active substances and a herbal medicine [1]. It is widely cultivated and consumed in China and is known as “Centennial ginseng” and “Mediterranean gold” [2]. Its underground tubers are rich in protein, polysaccharides, allantoin, vitamin C, calcium, iron, and other nutrients [3]. According to the traditional Chinese medicine theory, medicinal use belongs to the underground part of the medicinal herbs, which nourish the stomach, relieve fatigue, and lower blood sugar [4]. According to statistics, there are more than 600 yam species worldwide, and these yams are mainly distributed in regions such as Africa, Asia, and America. Among them, China is one of the wealthiest countries in terms of yam species, with 93 different species accounting for about one-sixth of the global yam species [5]. However, fresh yam has high crispness and moisture content (84%), making it prone to mechanical damage and moldy growth during sale, processing, transportation, and storage, leading to food waste and environmental pollution [6]. Drying techniques can reduce yams’ moisture content and water activity to extend their storage period and ensure their quality and nutrients [7].

Infrared combined hot air drying (IR-HAD) utilizes infrared radiation and hot air heat transfer drying. Infrared can penetrate the surface layer of the material and directly heat the internal moisture. The hot air propagation system accelerates the evaporation and discharge of moisture through the air supply cycle to improve the drying effect [8]. This process combines the excellent characteristics of infrared drying, fast warming speed, low energy consumption, and the advantages of hot air drying, such as low cost, easy operation, and suitability for large-scale production, which is widely used in sheet materials [9]. Feng et al. [10], by studying the effects of different drying methods on the quality of garlic flakes, found that the garlic flakes of IR-HAD have significant advantages in the rehydration capacity, the content of biological compounds, and the antioxidant properties. Xu et al. [11] studied the effects of different temperatures on the drying characteristics and quality of goldenseal. They concluded that far infrared combined with hot air drying shortened the drying time, and goldenseal color, flavor, and active ingredients were superior to those of far infrared drying. Zhang et al. [12] found that IR-HAD had the advantages of short drying time, low energy consumption, and high quality of dried cucurbits relative to hot air drying. It has a short drying time, low energy consumption, and good quality, which proves that IR-HAD is a suitable technology for maintaining the quality of fruits and vegetables. On the other hand, there are fewer studies in the literature reporting on IR-HAD for yam slices, and the technology is yet to be used to explore the effect of drying conditions on the pattern of physicochemical indices and the optimal process parameters.

Dry processing involves several indicators, such as efficiency, energy consumption, and quality, and it is not feasible to pursue only high efficiency or high quality; the key lies in selecting reasonable process parameters [13]. Most optimizations in food engineering involve single-objective optimization using response surface methodology (RSM), which has obvious advantages in terms of the amount of information obtained from evaluating the goodness of fit [14]. However, RSM analysis is usually based on linear models, and in non-linear problems, the results of RSM analysis may not accurately predict the output effects. In recent years, with the development of artificial intelligence, BP neural networks, as a data-driven tool, have demonstrated excellent adaptability in function approximation, non-linear function fitting, and online prediction, and they are one of the most widely used neural network models [15]. BP neural networks have been widely recognized as effective for solving complex, highly non-linear, and poorly defined science and engineering problems [16]. BP neural networks have been used for the prediction of hot air drying characteristics of wheat [17], microwave vacuum drying characteristics of carrots [18], prediction of polysaccharides in rhizomes of Atractylodis macrocephaly [19], prediction of flavor changes in the drying process of ginger [20], and prediction of infrared drying of broccoli moisture ratio [21]. Although BP neural networks have many advantages, they also have some limitations, such as easy overfitting, sensitivity to initial weight selection, and the need for a large amount of training data [22]. In practical applications, it is necessary to choose the appropriate network structure and learning algorithm according to the specific problem’s needs and the data’s characteristics and carry out appropriate tuning and optimization [23]. The gray wolf algorithm is based on media intelligence, which solves the optimization problem by simulating the way of interacting in the gray wolf group [24]. Based on the collaborative and competitive behaviors of gray wolves, the gray wolf algorithm uses the guiding role of the leader individual and the following behaviors of the other individuals to search for the optimal solution step by step through several iterations in order to obtain the global optimal solution for the problem [25]. In this regard, the BP neural network combined with the gray wolf algorithm is used to solve stochastic non-linear optimization problems to find the experiment’s optimal drying condition through fewer trials [26]. Most importantly, to the best of our knowledge, this is the first attempt to combine response surfaces with BP-GWO in the drying process, providing a new way of thinking about modeling and optimizing the drying process.

In this paper, the drying of yam slices was carried out in an infrared combined hot air drying oven to analyze its effect on the drying rate of yam slices. The effective water diffusion coefficient, unit energy consumption, color difference value, rehydration ratio, polysaccharide concentration, and allantoin concentration of yam slices were examined with various drying temperatures, slice thicknesses, and radiation distances. The optimal process parameters for IR-HAD of yam slices were predicted and optimized using BP-GWO based on RSM and then confirmed experimentally.

## 2. Materials and Methods

### 2.1. Materials

The Shihezi wholesale farmers’ market is where fresh yams were purchased. Selected yams 4 cm in diameter and 60 cm in length, devoid of disease spots, wounds, mold, and rot, were preserved in the refrigerator at 41 °C. Before drying, the sliced yams were washed with tap water to eliminate surface dirt and excess water. The yam was physically peeled and sliced into thin slices of the required thickness for the test, and numerous measurements were obtained for each sample using vernier calipers (precision of 0.02 mm) to eliminate yam slices with large inaccuracies. The original moisture content of the yam was assessed by drying the yams to a consistent mass in a hot air oven at 105 °C. The above test was repeated three times. The final average initial moisture content of yam was 84.28 ± 1.32% [27].

### 2.2. Test Method

This study used laboratory-scale integrated short- and medium-wave infrared and hot air combined drying equipment (STC et al., China; temperature accuracy ±0.1 °C, power range 0~2 kW) for the drying experiments. The internal dimensions of the drying chamber were 43 cm × 42 cm × 27 cm. Six infrared tubes were installed at the top to provide a heat supply with a wavelength range of 3~5 μm. During the drying process, the airflow was blown to the material tray through the blower, so that the material was dried uniformly. At the same time, a temperature sensor was placed on the material tray to measure the temperature in the drying chamber in time and control the temperature error within 0.1 °C.

### 2.3. Test Methods

To achieve optimal drying of yam slices, the infrared combined hot air dryer was adjusted to the desired temperatures (50 °C, 55 °C, 60 °C, 65 °C, and 70 °C) 30 min in advance. The wind speed was measured at the nozzle using an anemometer (TES-1340, Taiwan Tai-Style Electronics Industry Co., New Taipei City, Taiwan) at the center of the dryer, and the wind speed was adjusted to 3 m/s. Yam slices (150 g) of various thicknesses (2 mm, 4 mm, 6 mm, 8 mm, and 10 mm) were uniformly dispersed in a single layer on a 37 cm × 20 cm tray and placed in the respective radiation distances (80 mm, 120 mm, and 160 mm) for the drying experiments. The weight loss of the samples was evaluated using an electronic balance (BSM-4200.2, Shanghai JOYCE Electronics Co. Technology Co., Shanghai, China, sensitivity 0.01 g) every 15 min until the target moisture content (0.13 kg/kg) was reached. The samples were cooled in a cooling box (202-00A, Nanjing Changchuang Science and Technology Co., Ltd., Nanjing, China) and vacuum packed in low-density polyethylene (LDPE) bags for further analysis immediately after the test, and each set of tests was performed three times.

### 2.4. Drying Characteristics

The yam slices’ moisture ratio (MR) and drying rate (DR) were measured every 15 min during the drying process to obtain the MR-t and DR-Mt curves.

Equation (1) [28] can be used to calculate the dry basis moisture content of yam slices during drying.
(1)$$M_{t}=\frac{W_{t}-G}{G}$$
where *M_t_* is the dry basis moisture content at instant *t*, expressed in g/g d.b., and *W_t_* is the total mass of yam slices at any moment *t*, expressed in g. *G* denotes the sample’s dry mass in grams.

Equation (2) was used to calculate the moisture ratio *MR* at various times.
(2)$$MR=\frac{M_{t}}{M_{0}}$$
where *MR* is the moisture ratio; *M_t_* is the dry basis moisture content at time *t*, expressed in g/g d.b.; and *M*_0_ is the beginning moisture content, expressed in g/g d.b.

The *DR* of dried yam slices was estimated using Equation (3) [29].
(3)$$DR=\frac{M_{t1}-M_{t2}}{t_{1}-t_{2}}$$
where *DR* is the drying rate of yam slices, g/(g·min)^−1^; *M_t_*_1_ is the dry base moisture content at the moment of *t*_1_, in g/g; *M_t_*_2_ is the dry base moisture content at the moment of *t*_2_, in g/g; and *t* is the time, h.

### 2.5. Effective Moisture Diffusion Coefficient

According to Fick’s second diffusion equation [30], the effective moisture diffusion coefficient (*D_eff_*) can be described by the drying data of test samples based on the linear equation of ln *MR*-*t*, as shown in Equation (4).
(4)$$\ln MR=\ln\frac{8}{\pi^{2}}-\frac{\pi^{2}D_{eff}}{L^{2}}t$$
where *D_eff_* is the effective moisture diffusion coefficient of yam slices in m^2^/s; *L* is the yam slice thickness in m; and *t* is the drying time in s.

### 2.6. Determination of Color and Luster

The color parameters (*L**, *a**, *b**) of dried yam slices were measured using a colorimeter (SMY-2000SF, Beijing Sheng Ming Yang Technology Development Co., Ltd., Beijing, China). Using the following formula [31], the color difference value E between the dried treated group and fresh samples was computed.
(5)$$\Delta E=\sqrt{{\left(L-L^{*}\right)}^{2}+{\left(a-a^{*}\right)}^{2}+{\left(b-b^{*}\right)}^{2}}$$
where *L*, *a*, and *b* represent the brightness, red-green value, and yellow-blue value of fresh yam, and *L**, *a*, b** represent the brightness, red-green value, and yellow-blue value of dried yam slices, respectively.

### 2.7. Determination of Rehydration Ratio

The beaker containing distilled water was placed in a 40 °C water bath. When the temperature of distilled water had stabilized, 5 g of yam slices was added to 50 mL of the liquid. They were removed after 2 h of immersion, and the surface was dried with absorbent paper and weighed on an analytical balance (BSM-4200.2, Zhejiang Nader Scientific Instrument Co., Ltd., Hangzhou, China) [32]. The formula for the rehydration ratio was as follows:(6)$$R_{r}=\frac{m_{1}}{m_{2}}R_{r}=\frac{m_{1}}{m_{2}}$$
where *R_r_* is the rehydration ratio in grams per gram; *m*_1_ is the mass of the sample after rehydration in grams; and *m*_2_ is the mass of the sample before rehydration in grams.

### 2.8. Microstructure

One of the three groups of sliced yam samples was selected as a typical sample, which was broken immediately after liquid nitrogen quick-frozen food processing treatment to produce a natural crunchy longitudinal section. The specimen was glued to a sample tray using a low-carbon conductive adhesive and metal sprayed to allow scanning via scanning electronic microscopy (SEM) [33]. Representative images were selected to be recorded and analyzed in depth by repeatedly studying the compositional configuration of each block.

### 2.9. Polysaccharide Content

Determination of yam polysaccharides [34]: 2 mL of each glucose standard solution (0.00 g/mL, 0.20 g/mL, 0.40 g/mL, 0.60 g/mL, 0.80 g/mL, and 1.00 g/mL) was taken into a cuvette. The standard curve of glucose solution measured by UV spectrophotometer (UV-1900i, Shimadzu Instruments Co., Ltd., Kyoto, Japan) at 490 nm was *y* = 11.8557*x* − 0.0206, *R*^2^ = 0.9564. Six grams of phenol (108-95-2, Wuhan Kemik Bio-medical Technology Co., Ltd., Wuhan, China) and ninety-four grams of water were put into a 250 mL beaker. A 6% phenol solution was obtained after thorough mixing. In each pre-treatment test group, 3 g of dried yam products was accurately weighed, crushed, ground into powder, and then screened through a 60-mesh sieve to obtain a fine powder. The powders of the different pre-treated samples were put into 10 mL test tubes, 1 mL of 6% phenol solution was added to each test tube, and 5 mL of sulfuric acid (GB/T601-2016, Tianjin Yongsheng Fine Chemical Co., Ltd., Tianjin, China) was added quickly. After thorough mixing, the test tubes were put into a thermostatic water bath at 90 °C. After 15 min, the test tubes were taken out of the thermostatic water bath and put into a cold water bath for 5 min, then put into a centrifuge (L0-LX-H1850, Shanghai Li-Chen Bonsi Instrument Science and Technology Co., Ltd., Shanghai, China) for centrifugation (6000 r/min, 10 min); the precipitate was poured off, and the supernatant was retained in the cuvette and determined by ultraviolet spectrophotometer at 490 nm. The absorbance was measured at 490 nm, and the polysaccharide content was calculated based on the glucose standard curve.

### 2.10. Allantoin Content

UV-visible high-performance liquid chromatography was used to determine the allantoin concentration [35]. The yam sample was precisely weighed at 2 g, crushed and ground into powder, passed through a 60-mesh sieve, and dissolved in 10 mL of distilled water; 95% ethanol was added 10–15 times with sonication for 10 min at 4 °C; 10 mL of the sample was taken, centrifuged (7000 r/min) for 10 min, and allantoin was analyzed by liquid chromatography. HPLC column: 5 m, 250 4.6 mm; flow rate: 0.2 mL/min; UV wavelength: 200 nm; mobile phase: EtOH/CHCL_3_/H_2_O (0.5/0.012/100). The allantoin concentration was measured colorimetrically.

### 2.11. Unit Energy Consumption

The energy consumption was measured by an electric meter (DL333502, Deli Group Ltd., Ningbo, China), which was zeroed before the start of each test, and the reading of the meter at the end of the test indicated the energy consumed in this test [9].

The energy required to remove a water unit was determined using Equation (7).
(7)$$\varphi =\frac{Q}{M}$$
where *φ* is the unit energy consumption, kJ·h/kg; *Q* is the total energy consumption determined by the electric meter at the end of drying, kJ·h; and *M* is the dehydrated mass of yam slices at the conclusion of drying, kg.

### 2.12. Indicator Weights

The coefficient of variation approach was employed in this work to calculate the weights of drying time, color difference value, unit energy consumption, rehydration rate, polysaccharide content, and allantoin content of yam slices under various drying circumstances [36].

The coefficient of variation was calculated as Equation (8).
(8)$$V_{i}=\frac{\sigma_{i}}{x_{i}}$$

The weights of the indicators for each sample were calculated as Equation (9).
(9)$$W_{i}=\frac{V_{i}}{\sum_{i=1}^{n}V_{i}}$$
where *V_i_* is the *i*th indicator’s coefficient of variation; *W_i_* is the *i*th indicator’s weight; *i* is the ith indicator’s standard deviation; and *x_i_* is the ith indicator’s mean.

### 2.13. Response Surface Experimental Design

Drying temperature A (55~65 °C), slice thickness B (4~8 mm), and radiation distance C (80~160 mm) were used as the response factors, and drying time(Y_1_), ΔE(Y_2_), unit energy consumption(Y_3_), rehydration ratio(Y_4_), polysaccharide content(Y_5_), and allantoin content(Y_6_) were used as the response values. This study used a Box–Behnken design to conduct a three-level response surface test through drying temperature, slice thickness, and radiation distance. The regression equation between each response value and drying conditions was established using Design-Expert 12 software [37]. Table 1 demonstrates each test factor and its coding.

Equation (10) demonstrates that a second-order polynomial response surface model was employed to fit the experimental data and describe the relationship between each response and the independent variables [38].
(10)$$Y=\beta_{0}+\beta_{1}X_{1}+\beta_{2}X_{2}+\beta_{3}X_{3}+\beta_{11}{X_{1}}^{2}+\beta_{22}{X_{2}}^{2}+\beta_{33}{X_{3}}^{2}+\beta_{12}X_{1}X_{2}+\beta_{13}X_{1}X_{3}+\beta_{23}X_{2}X_{3}$$
where *Y* is the response function for each response variable (*Y*_1_–*Y*_4_). *β*_0_ is the constant term. *β*_1_, *β*_2_, and *β*_3_ are the linear coefficients. *β*_11_, *β*_22_, and *β*_33_ are the quadratic coefficients. *β*_12_, *β*_13_, and *β*_23_ are the interaction coefficients.

### 2.14. GWO-BP Model and Multi-Objective Optimization

BP neural network is a typical artificial neural network: a forward feedback network trained based on the backpropagation algorithm [39]. The BP neural network is composed of three layers—input, hidden, and output—with each node in the hidden and output levels coupled to the nodes in the previous layer [40]. A one-way test was used to determine the drying temperature, slice thickness, and radiation distance as input layer neurons of the BP neural network, and drying time, E, unit energy consumption, rehydration ratio, polysaccharide content, and allantoin content as output layer neurons. The number of nodes in the hidden layer was calculated using Equation (11) to construct the network structure of 3-m-6; 80% of the response surface data was selected for training, and 20% was used for testing and verification to complete the self-learning training and prediction of BP neural network. The specific BP neural network operation flow is shown in Figure 1.
(11)$$M=\sqrt{N+L}+a$$
where *L* is the number of nodes in the output layer; *M* is the number of nodes in the hidden layer; *a* is a constant of [0,10]; *N* is the number of nodes in the input layer.

Because of the significant differences in the scale and reference ranges of the judgment indicators (drying time, Δ*E*, energy consumption per unit, rehydration ratio, polysaccharide content, and allantoin content), the objective function was defined as follows:(12)$$\min f=\sum w_{i}{\left(1-k_{i}\right)}^{2}$$
where *min* is the minimum value of *f* taken; *i* is the index number; *w_i_* is the weight of the *i*th indicator; *k_i_* is the correlation between the calculated value of the *i*th response surface regression equation fact and the optimal knowledge of the best indicator.

For the smaller and better indicators *k_i_*_1_ (drying time, color difference value, and unit energy consumption) and the larger and better indicators *k_i_*_2_ (rehydration ratio, polysaccharide content, and allantoin content), the calculation formulae are shown in Equations (13) and (14), respectively.
(13)$$k_{i1}=\frac{f_{best}}{f_{act}}$$
(14)$$k_{i2}=\frac{f_{act}}{f_{best}}$$

The gray wolf algorithm has three phases: encirclement, hunting, and capture. The encirclement of prey is actually achieved by *α*, *β*, and *δ* wolves. The other wolves keep updating their positions through their distance from the positions of *α*, *β*, and *δ* wolves, which in turn capture the prey [41]. The position of each gray wolf iteration represents the solution set of the problem. The gray wolf will update its position according to Equations (15) and (16) for encirclement.

The iterative position and capture equations for the wolf position are as follows:(15)$$D_{i(k)}=\left|C\cdot X_{p(k)}-X_{i(k)}\right|$$
(16)$$X_{p(k+1)}=X_{p(k)}-A\cdot D$$
(17)$$A=2ar_{1}-a$$
(18)$$C=2r_{2}$$
where *D_i(k)_* is the distance of individual prey; *X_p(k)_* denotes the current position; *k* denotes the current number of iterations; *A* and *C* denote the coefficients; *r*_1_ and *r*_2_ denote the random vectors [0,1]; and *X*_*p*(*k*+1)_ denotes the position after the iteration. The elements of vector *a* decrease linearly from 2 to 0 with iterations.

The position update of the gray wolf hunting process is shown in Figure 2. The wolves gradually approach the prey according to the leadership mechanism, and each time, the position of the wolves is updated depending on the indication of Alpha, Beta, and Delta; the mathematical model equations are shown in Equations (19) and (20) [42].
(19)$$X(k+1)=\frac{1}{3}X_{1}+\frac{1}{3}X_{2}+\frac{1}{3}X_{3}$$
(20)$$\left\{\begin{array}{l} X_{1}=X_{\alpha }(k)-A_{1}\cdot D_{\alpha }\\ X_{2}=X_{\beta }(k)-A_{2}\cdot D_{\beta }\\ X_{3}=X_{\delta }(k)-A_{3}\cdot D_{\delta } \end{array}\right.\;\begin{array}{l} D_{\alpha }=\left|C_{1}\cdot X_{\alpha }-X\right|\\ D_{\beta }=\left|C_{2}\cdot X_{\beta }-X\right|\\ D_{\delta }=\left|C_{3}\cdot X_{\delta }-X\right| \end{array}$$

The main difference between the multi-objective gray wolf algorithm and the GWO algorithm is the introduction of the Archive population to change the way of selecting *α* (optimal individual), *β* (suboptimal individual), and *δ* (worst individual) [43]. The excellent individuals (non-dominated solutions) will be stored in the Archive in each generation. The Archive will perform update and deletion operations according to a particular strategy. Eventually, the individuals selected in the Archive are considered to be the optimized problem solutions of the Pareto optimal solution. The flowchart of the gray wolf algorithm is shown in Figure 1.

The maximum values of the optimization objectives, as shown in Equation (21), are rehydration ratio, polysaccharide content, and allantoin content, and the minimum values of the optimization objectives are drying time, color difference, and unit energy consumption.
(21)$$\mathrm{objectives}\;\left\{\begin{array}{l} \mathrm{Min}\;Y_{1}\left(A,B,C\right);\;{\mathrm{Min}\;Y}_{2}\left(A,B,C\right);\;{\mathrm{Min}\;Y}_{3}\left(A,B,C\right)\\ {\mathrm{Max}\;Y}_{4}\left(A,B,C\right);\;{\mathrm{Max}\;Y}_{5}\left(A,B,C\right);\;{\mathrm{Max}\;Y}_{6}\left(A,B,C\right)\\ 55\;{}^{\circ}C<A<65\;{}^{\circ}C;\;4\;\mathrm{mm}<B<8\;\mathrm{mm};\;80\;\mathrm{mm}<C<160\;\mathrm{mm} \end{array}\right.$$

All codes in this paper were written in a MATLAB environment, with the population size set at 100 and the maximum selection of external archives at 100 for a total of 1000 iterations. The procedure was run on a computer configured with Intel(R) Core(TM) i7, with 16 GB of onboard RAM, a 64-bit operating system, and a x64 based processor.

The coefficient of determination, *R*^2^, and the root mean square error, *RMSE* [36], were used to compare the model’s applicability to the data. The closer the *R*^2^ is to 1 and the lower the *RMSE* value, the more applicable the model. The calculations are performed in Equations (22) and (23).
(22)$$R^{2}=1-\frac{{\sum_{i=1}^{N}\left(y_{act,i}-y_{pre,i}\right)}^{2}}{\sum_{i=1}^{N}{\left(y_{act,i}-y_{i}\right)}^{2}}$$
(23)$$RMSE={\left[\frac{1}{N}{\sum_{i=1}^{N}\left(y_{act,i}-y_{pre,i}\right)}^{2}\right]}^{\frac{1}{2}}$$
where *y_act,i_* is the sample’s measured value; *y_pre,i_* is the sample’s predicted value; and *y_i_* is the average of the actual values. *M* and *n* are the number of data sets measured in the experiment and, accordingly, the number of constants in the model.

### 2.15. Data Processing

The experimental data were processed in Excel 2021 and plotted in Origin 2021. Response surface analysis was performed in Design-Expert 12. Modeling and optimization of BP-GWO were achieved in MATLAB R2018b.

## 3. Results and Analysis

### 3.1. Effect of Different Temperatures on Drying Characteristics and Quality of Yam Slices

#### 3.1.1. Effect of Different Temperatures on Drying Characteristics

Figure 3a shows the changes in yam slices with drying time under different drying temperature conditions. The drying moisture content showed an exponential decreasing trend, and the decreasing trend of the drying moisture content ratio of yam slices during the drying process was faster at the initial stage, slower in the middle stage, and it continued to slow down until the decreasing trend was flat. In addition, the higher the drying temperature, the steeper the curve, i.e., the higher the drying temperature, the shorter the drying time of yam slices. When the drying temperature was 70 °C, it only took 165 min for the yam slices to reach equilibrium, which was 38.9% shorter than that under 50 °C [44]. According to Figure 3b, it can be seen that different drying temperatures had a significant effect on the drying rate of yam slices, and higher drying temperatures could enhance the drying rate (*DR*) of yam slices. At the initial drying stage, the drying rate of yam slices was at its maximum. However, as the drying process of yam slices continued, the surface where water evaporated continuously migrated to the interior, and the distance of water diffusion increased. The resistance of water diffusion from the interior of yam slices to the surface was more excellent than that of water diffusion from the surface of yam slices to the air, which led to a gradual decrease in the drying rate [45].

#### 3.1.2. Effect of Different Temperatures on Effective Water Diffusion Coefficient of Yam Slices

The linear regression equations and effective moisture diffusion coefficients at various temperatures are shown in Table 2. The effective moisture diffusion coefficients were 8.5510^−9^, 9.8910^−8^, 1.1810^−8^, 1.3210^−8^, and 1.7910^−8^ when the drying temperatures were 50, 55, 60, 65, and 70 °C, respectively, and the effective moisture diffusion coefficients rose 2.09 times with temperature, showing that high temperature can provide more energy to enhance water diffusion from the interior to the outside and that increasing the temperature can increase the rate of water diffusion in yam slices. Zhang et al. [46] reached the same conclusion when drying yam slices in infrared hot air.

#### 3.1.3. Effect of Different Temperatures on Unit Energy Consumption of Yam Slices

The unit energy consumption reflects the power consumption of the infrared combined hot air dryer for yam slices. Figure 4 demonstrates the comparison of unit energy consumption at different drying temperatures. It was found that the lower the temperature, the greater the unit dewatering energy consumption, and the unit dewatering energy consumption of yam slices under the drying condition of 70 °C was reduced by 6.51, 5.08, 3.08, and 0.71 kW·h/kg compared with that under the drying condition of 50, 55, 60, and 65 °C, respectively. This was due to the longer drying time and the prolonged operation of the moisture removal fan, the heating film, and other electrical components consuming electric energy, and the unit dewatering energy consumption positively correlated with the drying time. The energy consumption per water removal unit and drying time were also positively correlated.

#### 3.1.4. The Effect of Different Temperatures on the Rehydration Ratio and Microstructure of Yam Slices

The rehydration ratio is an essential index for evaluating drying quality because it can characterize the degree of drying destruction in the material’s structure; the higher the rehydration ratio, the better the product quality and the less drying destruction in the product’s structure [47]. Figure 4 depicts the rehydration ratio of yam slices using IR-HAD at various temperatures [46]. As seen in Figure 4, the rehydration ratio of yam slices increased initially and then reduced with increasing temperature. The drying temperature of 60 °C resulted in the highest rehydration ratio of yam slices. When the drying temperature was raised to 70 °C, the rehydration ratio of yam slices was found to be the lowest. This may be because, as the drying temperature increases, more energy is absorbed per unit mass of yam slices, which generates a higher vapor pressure inside its tissues, leading to an increase in the expansion of its internal tissues and an increase in the rehydration ratio. However, a drying rate which was too fast caused rapid evaporation of water from the surface of the yam slices, resulting in the formation of a hard shell on the surface [48]. Under these conditions, the water inside the yam slices could not evaporate sufficiently, resulting in the hardening of the surface structure, which in turn led to a decrease in the rehydration ratio. In addition, the scanning electron microscopy observations in Figure 5 showed that different drying temperatures were essential factors affecting the microstructure of yam slices. With the increase in drying temperature, the porosity of yam slices showed a tendency to increase and then decrease. At 60 °C, the pores were most uniformly arranged, with a loose structure and the highest porosity. This may be because increasing the temperature will make the internal organization of yam slices more expansive, which is conducive to forming pores. However, too high a temperature will lead to its hardening [49]. The trend of rehydration ratio can also be demonstrated in the changes in microstructure.

#### 3.1.5. Effects of Different Temperatures on Color Parameters of Yam Slices

The color parameters of yam slices after infrared combined hot air drying at different drying temperatures are shown in Figure 6. When the drying temperature increased, the color of yam slices showed a decreasing change trend, and there was a significant difference between temperature and brightness *L** (*p* < 0.05). The brightness of yam slices was highest at a drying temperature of 50 °C, which was closest to that of fresh yam slices, probably due to the lower drying temperature, which reduced the possibility of destructive occurrence of nutrients at high temperatures. The brightness decreased with increasing temperature, which may be due to the effect of oxidative browning, which deteriorated the color. Δ*E* was greatest at 70 °C, which was 2.1 times higher than that at 50 °C. When the temperature rises, the movement among the molecules accelerates, resulting in a higher breakdown of pigments, enzymes, and other chemicals in the yam slices, intensifying the color of the slices. According to Zheng et al. [50], lower temperatures can decrease the color difference value of dried items.

#### 3.1.6. Effect of Different Temperatures on Polysaccharide Content and Allantoin Content of Yam Slices

Polysaccharide is a heat-sensitive substance; as one of the main active ingredients of yam, it exerts hypoglycemic, antioxidant, antitumor, and immune enhancement effects [34]. The effect of IR-HAD on polysaccharides of yam at different temperatures is shown in Figure 7. As seen in the figure, the drying temperature significantly affects yam polysaccharides. The polysaccharide content of fresh yam was 36.72 mg/g. Compared with the fresh samples, the polysaccharide content decreased by 29.23%, 26.48%, 23.35%, 45.76%, and 49.7% after drying at 50 °C, 55 °C, 60 °C, 65 °C, and 70 °C with IR-HAD, respectively. The highest polysaccharide content was found at 60 °C, which may be due to the appropriate drying time; a long drying time did not lead to oxidative decomposition of polysaccharide substances, and high-temperature drying did not lead to degradation of polysaccharide substances. Zheng et al. [50], by studying the polysaccharide content of winter wheat, found that when the drying temperature was within the range of 50–70 °C, the polysaccharide content of winter wheat first increased and then decreased. The content was highest at 60 °C, confirming polysaccharides’ properties.

Allantoin has anti-inflammatory, antioxidant, keratolytic, and skin-softening effects [51]. The allantoin content of yam slices treated at different temperatures ranged from 1.93 to 2.66 g/g. The loss of allantoin content in yam slices relative to fresh samples after drying at 60 °C was 10.24%; it was 22.41% after drying at 65 °C and 34.17% after drying at 70 °C, which may be due to the rapid degradation of allantoin substances at high temperatures [52], and the loss of allantoin content in yam slices after drying at 50 °C may also be attributed to the rapid degradation of allantoin substances due to prolonged contact with oxygen [53]. Prolonged contact with oxygen may have also contributed to the depletion of allantoin components.

### 3.2. Effects of Different Slice Thicknesses on Drying Characteristics and Quality of Yam Slices

#### 3.2.1. Effect of Different Slice Thicknesses on Drying Characteristics

Figure 8 shows the drying characteristics and drying rate curves of yam slices with different slice thicknesses when the drying temperature and radiation distance were 60 °C and 120 mm. The time required to dehydrate yam slices increased with increasing slice thickness, as shown in Figure 8a. The drying time of yam slices was 135, 165, 195, 285, and 360 min at the thickness of 2, 4, 6, 8, and 10 mm, respectively; the drying time increased by about 62.5% when the thickness of the slices was increased from 2 mm to 10 mm. Figure 8b shows that the drying rate of the yam slices increased briefly and then decreased slowly with the decrease in the moisture content of the dry base, which was basically in the stage of descending drying rate. This is because, in the initial drying stage, the moisture on the surface of the yam slices rapidly evaporates to the surrounding hot air under a large temperature and humidity gradient. The drying rate increases rapidly, accompanied by a decrease in the temperature gradient between the yam slices and the surrounding environment, and the drying rate gradually decreases [53]. In addition, the thicker the yam slices, the longer it takes for the internal moisture to migrate to the surface; thus, the drying rate decreases with increasing thickness of the slices. This trend is similar to the results of Zhang et al. [54], who reported far infrared drying of ginger slices in their study.

#### 3.2.2. Effect of Different Slice Thicknesses on the Effective Water Diffusion Coefficient of Yam Slices

Table 3 shows the effective moisture diffusion coefficients of yam slices under different slice thicknesses when the drying temperature and radiation distance were 60 °C and 120 mm. When the thickness of the slices was 2, 4, 6, 8, and 10 mm, the effective moisture diffusion coefficients were 1.79 × 10^−8^, 1.43 × 10^−8^, 1.18 × 10^−8^, 7.88 × 10^−9^, and 5.92 × 10^−9^, respectively, and the effective moisture diffusion coefficients decreased with the increase in the slice thickness. This was because the increase in slice thickness prolonged the transport path of water molecules, weakened the diffusion and migration of water molecules, and decreased the effective water diffusion coefficient [53]. Therefore, reducing the slice thickness is favorable for shortening the drying time and energy consumption.

#### 3.2.3. Effect of Different Slice Thicknesses on Unit Energy Consumption of Yam Slices

Figure 9 demonstrates the comparison of unit energy consumption according to yam slice thickness. It can be seen in the figure that the thicker the slice thickness, the greater the unit dewatering energy consumption, and the unit dewatering energy consumption of yam slices under the condition of 10 mm increases by 70.56%, 58.48%, 45.43%, and 27.67% compared to that under the conditions of 2, 4, 6, and 8 mm, respectively. Therefore, reducing the thickness of the slices is beneficial for shortening the drying time and reducing energy consumption.

#### 3.2.4. Effect of Different Slice Thicknesses on Rehydration Ratio of Yam Slices

Figure 9 shows the effect of different slice thicknesses on the rehydration ratio of yam slices. The figure shows that the rehydration ratio decreases with the increase in slice thickness. The lowest rehydration ratio was found at a slice thickness of 10 mm, possibly because thicker yam slices mean that the heat needs to pass through more material to reach the center portion. This caused the center to be heated slowly and dried unevenly, resulting in over-drying of the outer layer, while the center still contained water. Over-drying of the outer layer may lead to cell wall disruption and structural hardening, which reduces the ability of water to enter the yam slices after drying [55]. Figure 10 shows the effect on the microstructure of yam slices at different slice thicknesses. It can be seen in the figure that the microstructure’s pore-like structure gradually increased with the thickness of the slices, and the pore-like structure was most clearly visible at 2 mm. The reasons may be that thin slices are more quickly and evenly heated; the heat can be rapidly transferred to each part of the yam slices; the moisture needs to pass through a shorter distance; the migration rate is faster; and it is easy to form a large number of delicate pores [55]. On the contrary, the temperature difference between the interior and exterior of thicker slices may be more significant, resulting in insufficient internal heat transfer and a lower rate of moisture migration, and the process of moisture evaporation may lead to contraction of the internal structure, and thus, a reduction in the number and size of holes.

#### 3.2.5. Effect of Different Slice Thicknesses on Color Parameters of Yam Slices

The color parameters of yam slices after infrared combined hot air drying at different slice thicknesses are shown in Figure 11. There was a significant difference (*p* < 0.05) in the effect of the thickness of yam slices on brightness *L** and Δ*E*. *L* decreased with the increase in slice thickness, and E increased with the increase in slice thickness. The maximum value was 2 mm, and a minimum color difference was found. This is because the pigments in the yam slices may undergo degradation or oxidation reactions during drying, and these reactions may intensify with the time and temperature of heat treatment. Since the internal temperature of thicker slices rises more slowly, they may be subjected to heat for a more extended period overall during the drying process, which may lead to color changes and color differences [56].

#### 3.2.6. Effect of Different Slice Thicknesses on Polysaccharide Content and Allantoin Content of Yam Slices

Figure 12 shows the effect of different slice thicknesses on the polysaccharide content of yam. As can be seen in the figure, the thickness of yam slices had a significant effect on polysaccharide content. The polysaccharide content increased and then decreased with the increase in slice thickness, and the highest polysaccharide content was found at 6 mm. It is possible that when the slices are thin (e.g., 2 mm), the drying rate of yam slices is faster, and the water evaporates rapidly, resulting in excessive heat exposure and degradation loss of the heat-sensitive polysaccharide components [57]. As the thickness of the slices increases (to the middle range, e.g., 6 mm), the heat transfer may be more uniform, and the rate of water evaporation inside the yam is balanced with heat absorption, which is favorable for the preservation of polysaccharides, while the mild drying conditions may reduce the degradation of heat-sensitive components [58]. When the thickness of yam slices is large (e.g., 10 mm), the outer layer’s rapid drying and the inner layer’s relative moistness can result in the migration of moisture from the inner layer of yam slices to the dry outer layer. During this process, soluble solids such as polysaccharides may also move with moisture to the outer layer and accumulate in this region. Since the temperature of the outer layer may be higher, or the drying more intense, these migrated polysaccharides are susceptible to thermal degradation or thermal damage, reducing the final measured polysaccharide content [59].

The effect of different slice thicknesses on the allantoin content of yam is shown in Figure 12. The allantoin content decreased with increasing slice thickness. The allantoin content decreased by 18.28%, 23.14%, 24.01%, 35.14%, and 50.29% after drying at 2, 4, 6, 8, and 10 mm thicknesses, respectively, compared to fresh samples. The most minor decrease in allantoin content was observed at 2 mm. This is because yam slices with a lower thickness were heated uniformly, and the drying time was shorter. Thicker yam slices require a longer time to achieve the same degree of drying, and more prolonged exposure to high temperatures during drying may lead to degradation of heat-sensitive components, such as allantoin [51].

### 3.3. Effects of Different Radiation Distances on Drying Characteristics and Quality of Yam Slices

#### 3.3.1. Effect of Different Radiation Distances on Drying Characteristics

The drying characteristics and drying rate curves of yam slices under different radiation distances are shown in Figure 13; the drying time of yam slices increased with the increase in radiation distance. When the radiation distance was 80 mm, the time for drying to reach equilibrium was the shortest, 40.12% shorter than the drying time when the radiation distance was 160 mm. The drying rate of the yam slices decreased with the increase in radiation distance, and with the increasing radiation distance, the drying energy of infrared obtained per unit mass of yam slices decreased, and thus, the drying rate decreased, resulting in prolongation of the drying time.

#### 3.3.2. Effect of Different Radiation Distances on Effective Water Diffusion Coefficient of Yam Slices

Table 4 shows the effective moisture diffusion coefficients of yam slices under different radiation distances when the drying temperature and slice thickness were 60 °C and 6 mm, respectively. When the radiation distance was 80, 120, and 160 mm, the effective moisture diffusion coefficients were 1.56 × 10^−8^, 1.18 × 10^−8^, and 1.07 × 10^−8^, respectively, and the effective moisture diffusion coefficients decreased with the increase in radiation distance. This is because the increase in radiation distance caused a higher loss of IR amount in the environment.

#### 3.3.3. Effect of Different Radiation Distances on Energy Consumption per Unit of Yam Slices

Figure 14 compares the energy consumption per unit of yam slices at different radiation distances. As can be seen in the Figure 14, the smaller the radiation distance, the greater the unit dewatering energy consumption. Under the same conditions, a smaller radiation distance can shorten the drying time to reduce energy loss.

#### 3.3.4. Effect of Different Radiation Distances on Rehydration Ratio of Yam Slices

Figure 14 shows the rehydration ratio of yam slices at different radiation distances. With the increase in radiation distance, the rehydration ratio tended to increase and decrease, and the highest rehydration ratio was found at the radiation distance of 120 mm. This may be because when the radiation distance was closer (e.g., 80 mm), the energy transfer to the yam slices was more efficient, resulting in rapid drying of the exterior of the yam slices and the formation of a more rigid outer skin layer, which may have limited the rapid evaporation of water from the interior [48]. When the radiation distance was about 120 mm, it may have provided just the right kind of temperature and drying rate, which enabled the removal of water from the inside and outside of the yam slices in a relatively uniform manner, with a moderate texture after drying, which was not overly complex or solid and did not affect the rehydration ratio due to insufficient drying. As the radiation distance increases, the energy radiated to the material will be dispersed due to the increase in propagation distance. The energy density will be reduced, weakening the heating effect, which may lead to the water in the center part of the yam slices not being able to evaporate sufficiently, thus affecting the increase in the rehydration ratio [60]. Figure 15 shows the microstructure of yam slices at different radiation distances. When the radiation distance is 80 mm, the infrared radiation heat is more significant, and the heat absorbed on the surface of the yam slices rapidly raises its surface temperature, which leads to the rapid evaporation of the surface water. However, due to the rapid evaporation of the surface water, a hardened layer will be formed, which prevents the evaporation of the internal water, resulting in an imperfect internal pore structure [48]. When the radiation distance is 120 mm, the intensity of infrared radiation may be just right to dry the yam slices uniformly without causing a hardened layer on the surface. Such distance and heat can promote uniform water evaporation from the inside, while the yam slices have time to form a good pore structure and achieve superior drying conditions. When the radiation distance is further increased to 160 mm, the intensity of infrared radiation is weakened, resulting in reduced heat transfer, insufficient heating of the surface and interior of the yam slices, and a slowdown in water evaporation; moreover the formation of a pore structure in the drying process may be hampered due to reduction in the water migration rate [49].

#### 3.3.5. Effects of Different Radiation Distances on Color Parameters of Yam Slices

The color parameters of yam slices after infrared combined hot air drying at different radiation distances are shown in Figure 16. Different radiation distances significantly affected *L** and color differences (*p* < 0.05). L decreased with the increase in radiation distance, and Δ*E* increased with the increase in radiation distance. The maximum value of L and minimum color difference were observed at 80 mm. As the distance between the IR source and the yam slices increases, the distribution of heat energy may become less uniform, resulting in inconsistent drying speed and degree of drying in various parts of the yam slices, and hence, the color difference. At closer distances (e.g., 80 mm), the heat distribution is more concentrated, which may contribute to faster and more uniform drying and less color shading. Secondly, a longer distance may mean less efficient heat transfer and slower response time, leading to lower drying results and more color unevenness [12].

#### 3.3.6. Effect of Different Radiation Distances on Polysaccharide Content and Allantoin Content of Yam Slices

Figure 17 shows the effect of different radiation distances on yam polysaccharide content and allantoin content. As can be seen in the figure, the radiation distance of yam slices had a significant effect on polysaccharide content and allantoin content. Both polysaccharide content and allantoin content increased and then decreased with increased radiation distance. A higher intensity of infrared radiation at a closer distance may lead to a sharp increase in the temperature on the surface of the yam when surface evaporation is intense. However, the internal water migration is slow, which can easily cause thermal degradation and loss of nutrients in yam slices. At an appropriate radiation distance (e.g., 120 mm), the drying rate and water migration may reach equilibrium, and the water evaporates uniformly, allowing better preservation of polysaccharides and allantoin [61]. Longer radiation distances result in longer drying times, meaning that the yam slices are exposed to oxygen and possible oxidative reactions for a more extended period, especially in the later drying stages when most of the moisture has been removed. The oxidative reactions may become more active, oxidizing polysaccharides and allantoin [62].

### 3.4. Response Surface Regression Model and ANOVA

There were 17 groups of test points in the response surface test, and the center test was repeated four times to estimate the test error. The specific yam slice IR-HAD combination drying response surface test results are shown in Table 5. Among them, the variation ranges of Y_1_, Y_2_, Y_3_, Y_4_, Y_5_, and Y_6_ were 150~255 min, 3.15~8.89, 6.74~12.24 kJ·h/kg, 16~26.23 mg/g, 2.14~2.96 kg/kg, and 2.46~3.71 μg/g, respectively.

The data in Table 5 were regressed using Design-Expert software, and the regression coefficients were tested for significance, as shown in Table 6. The Y_1_ regression model had a highly significant (*p* < 0.0001) effect on the primary term B, a significant (*p* < 0.05) effect on A, C, AC., and B.C., and a non-significant (*p* > 0.05) effect on AB., A^2^, B^2^, and C^2^. The Y_2_ regression models of A, B, and B^2^ were highly significant (*p* < 0.0001), and B and C^2^ were significant (*p* < 0.05). At the same time, AB., AC., BC, and A^2^ were insignificant (*p* > 0.05). The Y_3_ regression model of A, B, C, and A^2^ was significant *(p* < 0.05). At the same time, AB., A.C., BC, B^2^, and C^2^ were insignificant (*p* > 0.05). The Y_4_ regression model of A, B, C, A^2^, and B^2^ effects was significant (*p* < 0.05). In contrast, AB., AC., BC, and C^2^ effects were not significant (*p* > 0.05). The Y_5_ regression models of B, A^2^, and C^2^ effects were highly significant (*p* < 0.0001); A and C effects were significant (*p*< 0.05). In contrast, AB., AC., BC, and B^2^ effects were not significant (*p* > 0.05). The Y_6_ regression models of C, A^2^, and B^2^ were highly significant (*p* < 0.0001); A and B were significant (*p* < 0.05), while AB., AC., BC, and C^2^ were not significant (*p* > 0.05). The *R*^2^ and C.V. (%) for Y_1_, Y_2_, Y_3_, Y_4_, Y_5_, and Y_6_ were 0.9497, 0.9892, 0.9533, 0.926, 0.9767, and 0.9881 and 5.5%, 3.56%, 4.83%, 5.71%, 1.87%, and 2.01%, respectively, indicating that the model fit was high, the experimental reproducibility was good, and the results exhibited high accuracy. They can be used to analyze and predict the combined drying process of yam slices.

The simplified regression equations are shown in Equations (24)–(29), which retain the above model’s highly significant and significant terms and remove insignificant ones.
(24)$$Y_{1}=752.97794-13.875A+41.25B-4.17188C+0.09375AC-0.1875BC$$
(25)$$Y_{2}=12.00209+0.19375A-4.37902B-0.073732C+0.387001B^{2}+0.000194C^{2}$$
(26)$$Y_{3}=212.90483-7.1569A+0.368625B+0.027806C+0.061087A^{2}$$
(27)$$Y_{4}=-447.54243+15.50538A+6.04477B+0.029875C-0.132805A^{2}-0.556283B^{2}$$
(28)$$Y_{5}=-51.33316+1.77368A-0.095B+0.035428C-0.014947A^{2}-0.000143C^{2}$$
(29)$$Y_{6}=-62.2702+2.08336A+1.26194B-0.005375C-0.017584A^{2}-0.111151B^{2}$$

As can be seen in the variance results, the regression equations of the six indicators are significant (*p* < 0.05), and the lack of fit is not significant (*p* > 0.05), indicating that the experimental values and the regression model exhibit a good fit.

### 3.5. BP-GWO Modeling

For modeling a three-layer BP neural network, Lansing was chosen as the transfer function for the input layer, while Purelin was used as the transfer function for the output layer. In addition, training was used as the training function. After determining the number of nodes in the input and output layers, the number of nodes in the hidden layer significantly impacts the network’s performance. In order to achieve the best performance, the number of nodes from 3 to 13 in the hidden layer is selected. The corresponding *R*^2^ and *RMSE* are obtained, as shown in Figure 18. It can be seen in the Figure 18 that the number of neurons corresponding to the highest *R*^2^ and lowest *RMSE* for the six metrics (Y_1_, Y_2,_ Y_3_, Y_4_, Y_5_, and Y_6_) is found in group 8. The *R*^2^ and *RMSE* are 0.93051, 0.94232, 0.905, 0.94007, 0.95049, and 0.900015 and 11.5877, 0.45657, 0.56722, 0.67519, 0.066475, and 0.11764, respectively. The optimal network topology for this experiment is thus obtained as 4-8-1 and optimized by the gray wolf algorithm.

Single-objective optimization of the regression equation was carried out through Design Expert, and the results are shown in Table 7. In order to avoid subjective preferences affecting the judgment results, the weights of the six indicators were assigned by the coefficient of variation method, as shown in Table 8. The optimized values and weights in Table 8 and Table 9 were substituted into Equation (12) to obtain the optimized fitness function, respectively.
(30)$$\min f=0.18{\left(1-\frac{136.37}{Y_{1}}\right)}^{2}+0.25{\left(1-\frac{3.3}{Y_{2}}\right)}^{2}+0.16{\left(1-\frac{6.75}{Y_{3}}\right)}^{2}+0.15{\left(1-\frac{Y_{4}}{26.23}\right)}^{2}+0.12{\left(1-\frac{Y_{5}}{2.97}\right)}^{2}+0.13{\left(1-\frac{Y_{6}}{3.81}\right)}^{2}$$

A comparison of the BP neural network results optimized by the gray wolf algorithm with the response surface results is shown in Table 9. In the response surface model (RSM), the values of *R*^2^ range from 0.9502 to 0.9794, all of which are greater than 0.9501, while in the BP-GWO model, all values are greater than 0.9915. In addition, relative to the RSM, the RMSEs of the BP-GWO model are reduced by 61.34%, 47.04%, 80.03%, and 78.04% compared to 39.99% and 47.65% for the RSM, respectively. The better the fit of the model, the smaller the *RMSE*, the larger the *R*^2^ and the corresponding reduction in the fitting error. The comparison of the data shows that the *RMSE* of the BP-GWO model is significantly lower than the corresponding value of RSM, and the *R*^2^ of the BP-GWO model is higher than that of the RSM. Therefore, the BP-GWO neural network optimization model has a better fitting effect, making the prediction results more accurate. The six indicators (Y_1_, Y_2_, Y_3_, Y_4,_ Y_5_, and Y_6_) predicted with the RSM and BP-GWO models under 17 sets of test conditions are shown in Figure 19. The results show that the predicted values of the BP-GWO model are closer to the actual values than those of the RSM model, and the fit between the predicted and actual values is higher. Although the RSM model is widely used in process parameter optimization, it can only construct a second-order polynomial regression model; therefore, its application is limited. The BP-GWO model, on the other hand, can predict almost all forms of non-linearities and is therefore more appropriate, reliable, and accurate in prediction, among others [43].

### 3.6. Multi-Objective Optimization

Figure 20 shows the Pareto optimal set of 100 optimal solutions for the multi-objective gray wolf algorithm, and Figure 20A,B denote the lookahead small metrics (Y_1_, Y_2_, and Y_3_) and lookahead extensive metrics (Y_4_, Y_5_, and Y_6_), respectively. As can be seen in Figure 20A, the points in the Pareto optimal set gradually increase from the upper right corner to the lower left corner. The points inside the ellipse indicate that the points in the lower left corner are more optimal. In this optimal set, we can find the optimal values of 131.38 min, 7.18, and 8.45 kJ·h·kg^−1^ for Y_1_, Y_2_, and Y_3_, respectively. As seen in Figure 20B, the points in the Pareto optimal set gradually increase from the lower upper left corner to the upper right corner. The optimal solutions for the optimal Y_4_, Y_5_, and Y_6_ are found within the ellipse, with values of 21.08 mg/g, 2.95, and 3.78 μg/g, respectively. Given that each point represents the optimal case, in order to reduce the error of the optimal set, ten sets of optimal solution sets were selected from the ellipse in Figure 9 and averaged to obtain the optimal solutions for the six indices (Y_1_, Y_2_, Y_3_, Y_4_, Y_5_, and Y_6_) at the drying temperature, the thickness of the slices, and the radiation distance of 63.57 °C, 4.27 mm, and 91.39 mm, respectively. The optimum values were 133.71 min, 7.26, 8.54 kJ·h·kg^−1^, 20.73 mg/g, 2.84, and 3.69 μg/g, respectively, as shown in Table 10.

In order to determine the reliability of the results, they were validated against the actual conditions of the dryer, as shown in Table 11. Under the actual operation of drying temperature A = 64 °C, slice thickness B = 4 mm, and radiation distance C = 91 mm, the relative errors of the experimental and predicted values of Y_1_, Y_2_, Y_3_, Y_4_, Y_5_, and Y_6_ were 4.46%, 3.71%, 0.81%, 3.96%, 2.9%, and 1.65%, respectively, with the errors being less than 5%. The experimental values were close to the model’s predicted values, which indicated that the experimental process parameters optimized by the obtained model are reliable. Figure 21 and Figure 22 represent the microstructure and drying process of yam slices under the above conditions, where it can be seen that the optimized microstructure of yam slices had complete starch granules and a loose structure and appeared to have a porous channel structure. After the drying optimization process, the surface of the yam slices became uniform, with a slight golden yellow color, and it had a certain degree of contraction. In conclusion, optimization of the BP neural network using the gray wolf algorithm can effectively predict the process parameters of IR-HAD of yam slices, which shows the method’s feasibility.

## 4. Conclusions

In this paper, the effects of drying temperature (55–65 °C), slice thickness (4–8 mm), and radiation distance (80–120) on drying time (Y_1_), Δ*E* (Y_2_), energy consumption per unit (Y_3_), polysaccharide content (Y_4_), rehydration ratio (Y_5_), and allantoin content (Y_6_) of IR-HAD of yam slices were investigated using RSM and BP-GWO models. The results showed that drying temperature, slice thickness, and radiation distance had significant effects on Y_1_, Y_2_, Y_3_, Y_4_, Y_5_, and Y_6_ (*p* < 0.05). The *R*^2^ values of the models were all greater than 0.92, with a good fit and C.V. (%) ranging from 1.87% to 5.71%, with high accuracy. The *R*^2^ of the RSM model ranged from 0.9502 to 0.9794, while the *R*^2^ values of the BP-GWO model were all greater than 0.9915, with higher predictive accuracy and a reduction in RMSE from 61.34% to 80.03%. The optimal drying conditions obtained via multi-objective optimization of BP-GWO were A = 63.57 °C, B = 4.27 mm, C = 91.39 mm, and the optimal indices were Y_1_ = 133.71 min, Y_2_ = 7.26, Y_3_ = 8.54 kJ·h·kg^−1^, Y_4_ = 20.73 mg/g, Y_5_ = 2.84, Y_6_ = 3.69 μg/g. The relative errors between the actual and predicted values were verified by the test to be less than 5%, indicating that the process parameters optimized by the model were reliable. In summary, it is considered that the theories and techniques applied in this experiment are innovative and valuable for the optimization of the IR-HAD process of yam tablets.

## Figures and Tables

**Figure 1 foods-13-00434-f001:**
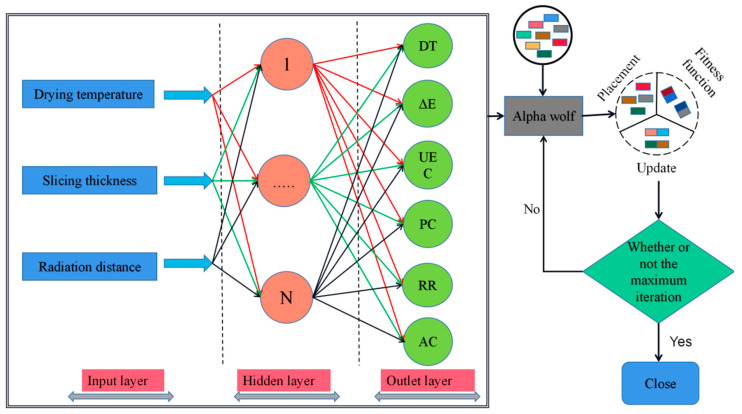
Flowchart of BP neural network and gray wolf algorithm.

**Figure 2 foods-13-00434-f002:**
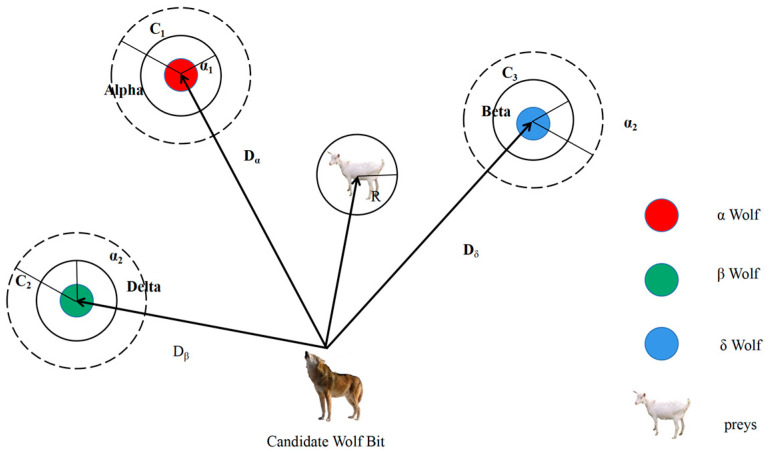
Schematic diagram of gray wolf location update.

**Figure 3 foods-13-00434-f003:**
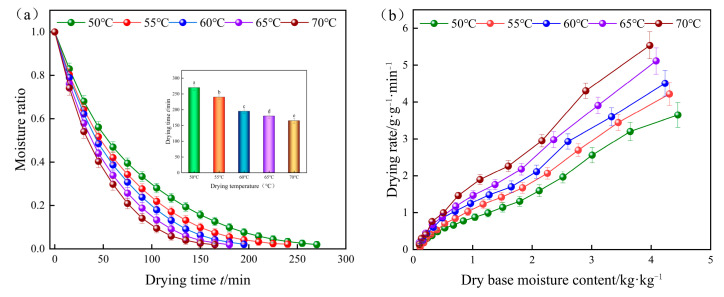
Water content (**a**) and drying rate (**b**) curves of yam slices at different drying temperatures. Note: Different letters in the graphs show significant differences according to the Duncan test (*p* < 0.05), as below.

**Figure 4 foods-13-00434-f004:**
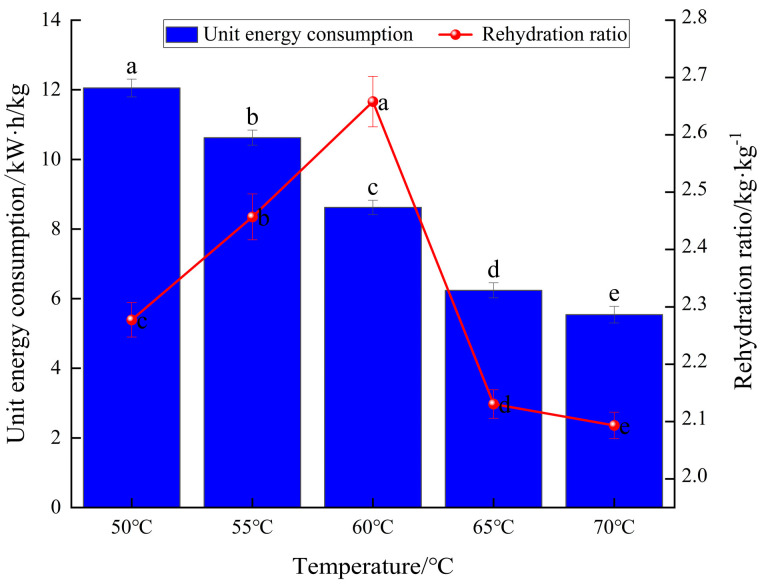
Effect of different drying temperatures on unit energy consumption and rehydration ratio of yam slices. Note: Different letters in the graphs show significant differences according to the Duncan test (*p* < 0.05), as below.

**Figure 5 foods-13-00434-f005:**
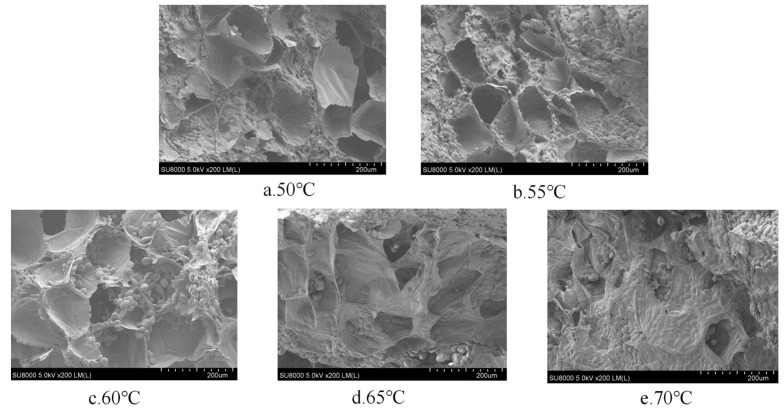
Effect of different drying temperatures on the microstructure of yam slices.

**Figure 6 foods-13-00434-f006:**
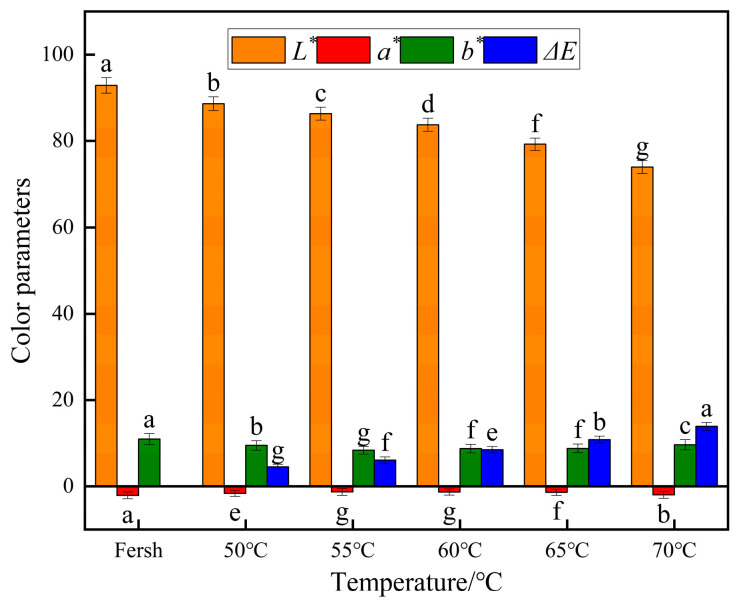
Effect of different drying temperatures on color parameters of yam slices. Note: Different letters in the graphs show significant differences according to the Duncan test (*p* < 0.05), as below.

**Figure 7 foods-13-00434-f007:**
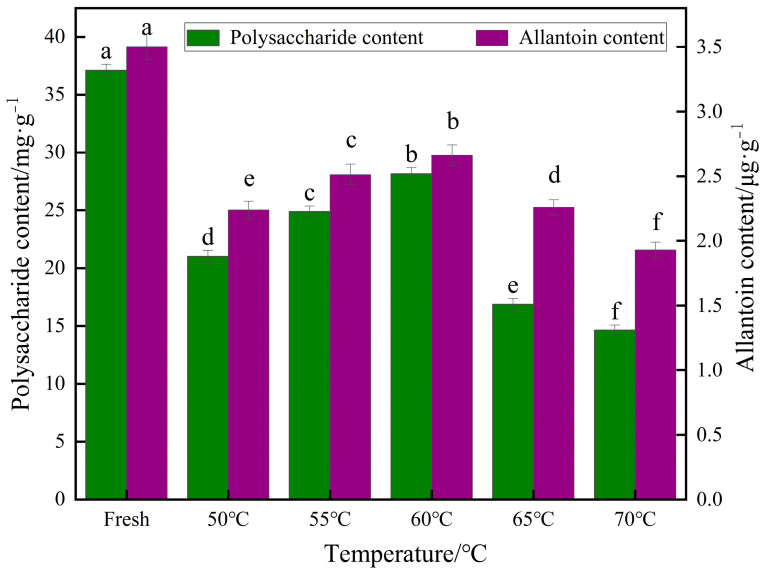
Effect of different drying temperatures on polysaccharide content and allantoin content of yam slices. Note: Different letters in the graphs show significant differences according to the Duncan test (*p* < 0.05), as below.

**Figure 8 foods-13-00434-f008:**
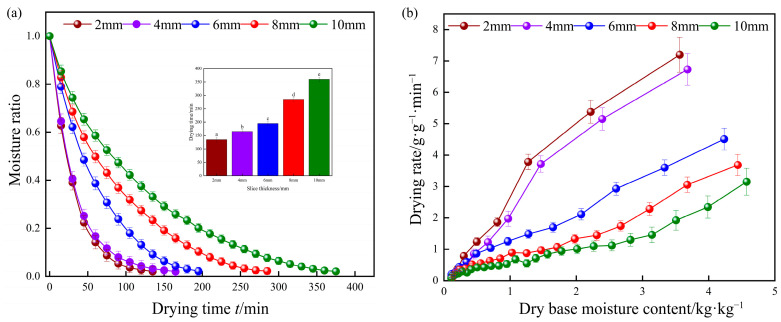
Curves of moisture content (**a**) and drying rate (**b**) of yam slices at different slice thicknesses. Note: Different letters in the graphs show significant differences according to the Duncan test (*p* < 0.05), as below.

**Figure 9 foods-13-00434-f009:**
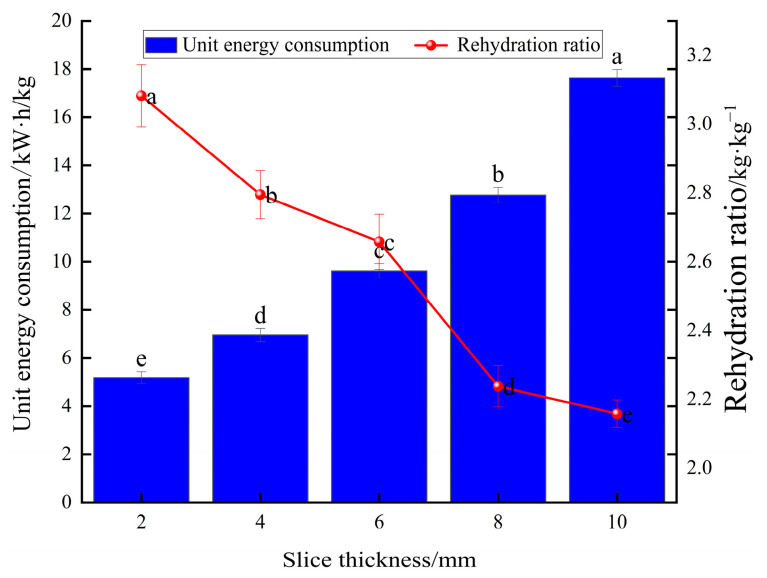
Effect of different slice thicknesses on unit energy consumption and rehydration ratio of yam slices. Note: Different letters in the graphs show significant differences according to the Duncan test (*p* < 0.05), as below.

**Figure 10 foods-13-00434-f010:**
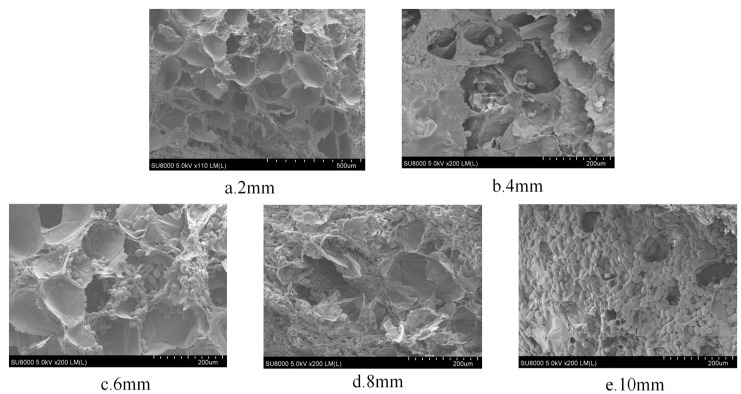
Effect of different slice thicknesses on the microstructure of yam slices.

**Figure 11 foods-13-00434-f011:**
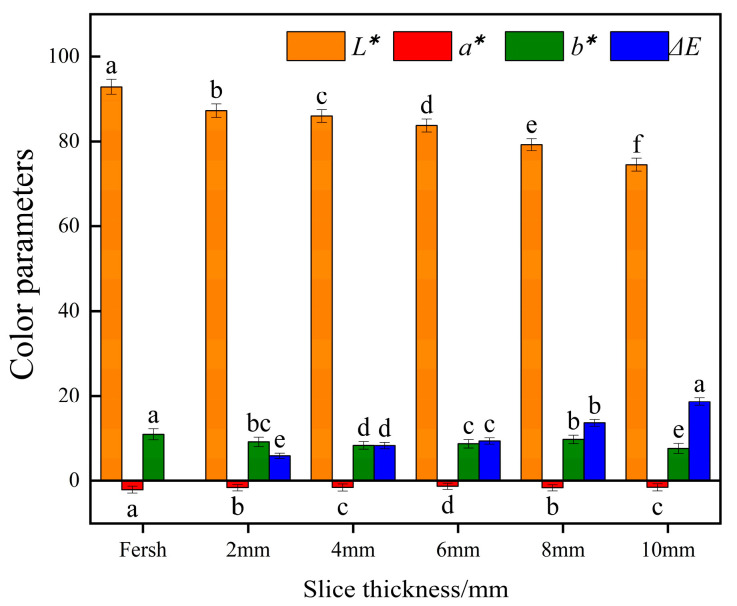
Effect of different slice thicknesses on color parameters of yam slices. Note: Different letters in the graphs show significant differences according to the Duncan test (*p* < 0.05), as below.

**Figure 12 foods-13-00434-f012:**
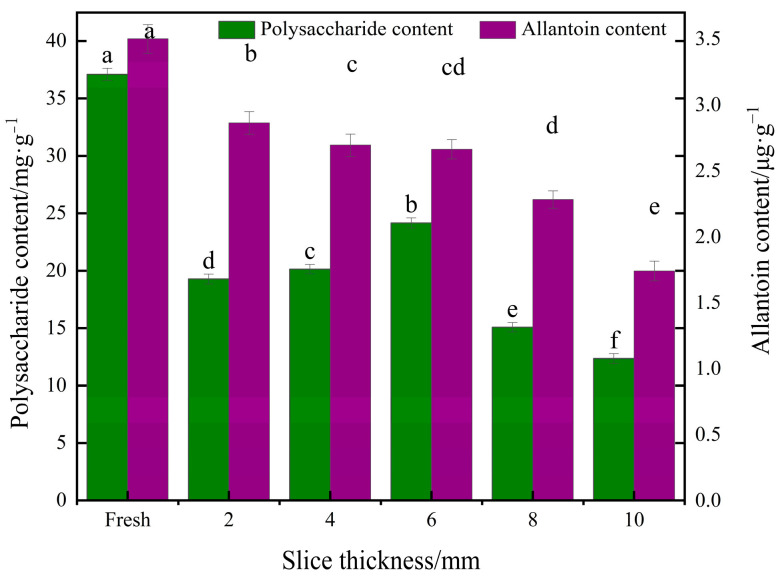
Effect of different slice thicknesses on polysaccharide content and allantoin content of yam slices. Note: Different letters in the graphs show significant differences according to the Duncan test (*p* < 0.05), as below.

**Figure 13 foods-13-00434-f013:**
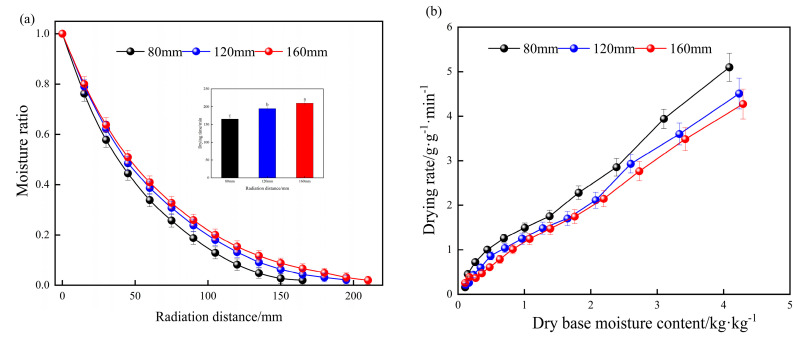
Curves of moisture content (**a**) and drying rate (**b**) of yam slices at different radiation distances. Note: Different letters in the graphs show significant differences according to the Duncan test (*p* < 0.05), as below.

**Figure 14 foods-13-00434-f014:**
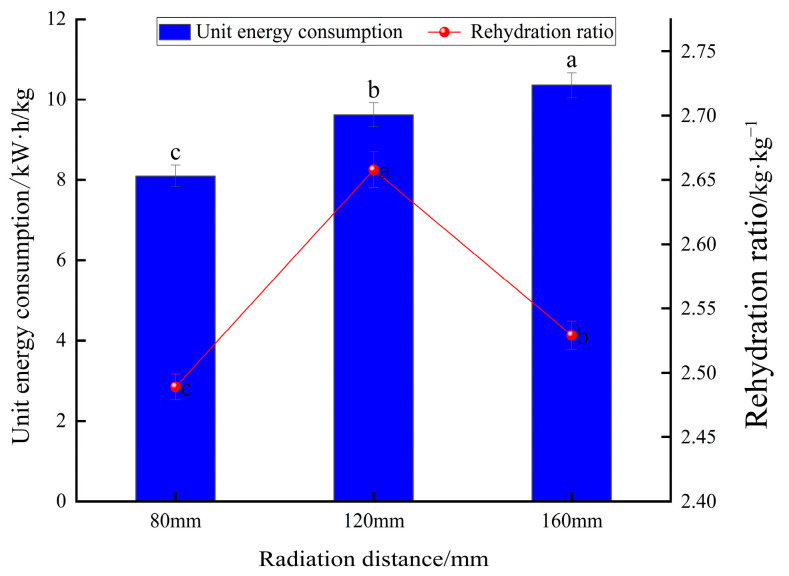
Effect of different radiation distances on unit energy consumption and rehydration ratio of yam slices. Note: Different letters in the graphs show significant differences according to the Duncan test (*p* < 0.05), as below.

**Figure 15 foods-13-00434-f015:**
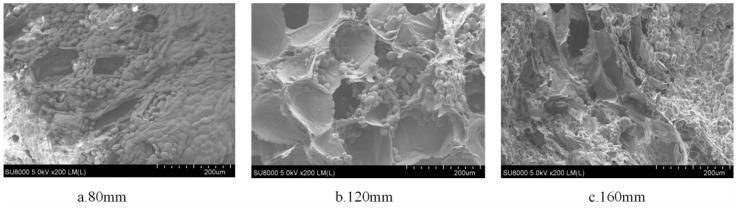
Effect of different radiation distances on the microstructure of yam slices.

**Figure 16 foods-13-00434-f016:**
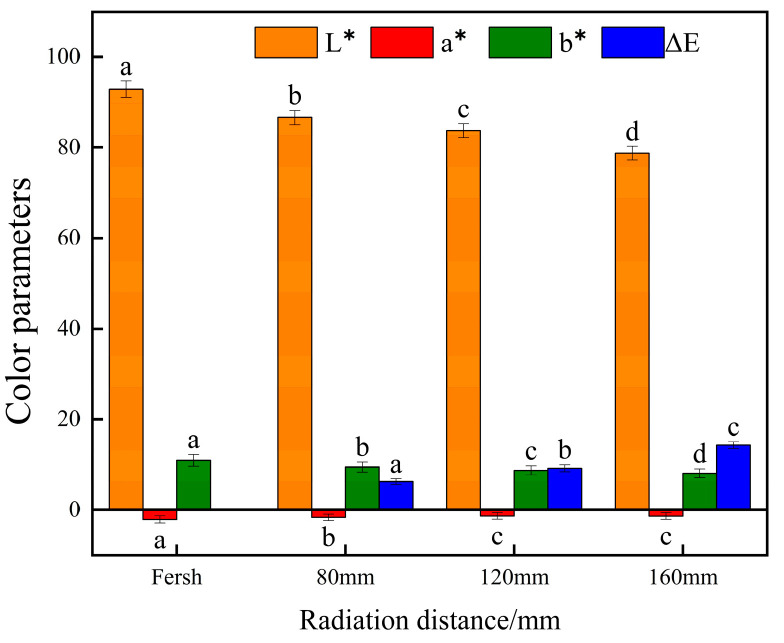
Effect of different radiation distances on color parameters of yam slices. Note: Different letters in the graphs show significant differences according to the Duncan test (*p* < 0.05), as below.

**Figure 17 foods-13-00434-f017:**
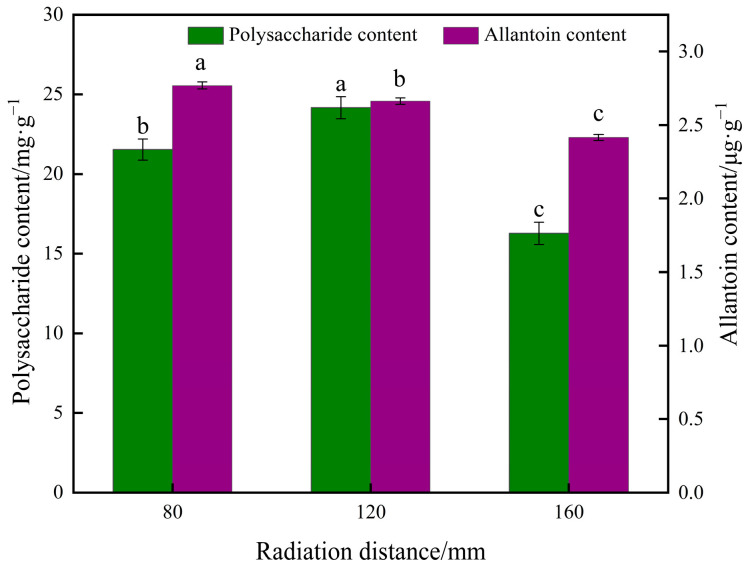
Effect of different radiation distances on polysaccharide content and allantoin content of yam slices. Note: Different letters in the graphs show significant differences according to the Duncan test (*p* < 0.05), as below.

**Figure 18 foods-13-00434-f018:**
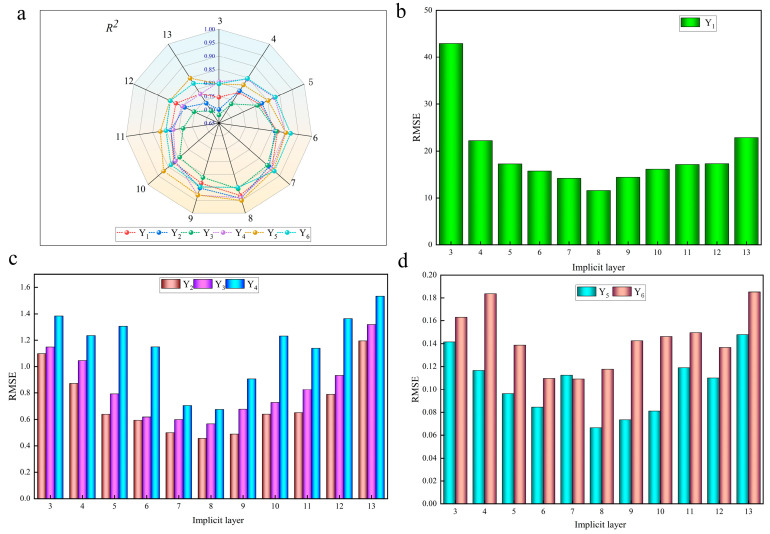
Simulation results of different hidden layers of BP neural network. Note: (**a**) shows the *R*^2^ corresponding to the implicit layers of Y1, Y2, Y3, Y4, Y5, Y6. (**b**–**d**) show, respectively, the RMSE of Y1, Y2, Y3, Y4, Y5, Y6.

**Figure 19 foods-13-00434-f019:**
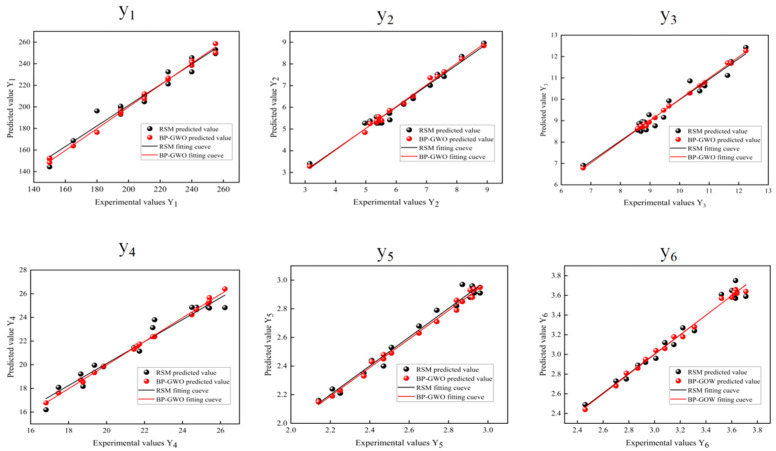
Linear fitting plots of predicted and actual values of response surface model and BP-GWO model. Drying time (y_1_); color difference value (y_2_); unit energy consumption (y_3_); polysaccharide content (y_4_); rehydration ratio (y_5_); and allantoin content (y_6_).

**Figure 20 foods-13-00434-f020:**
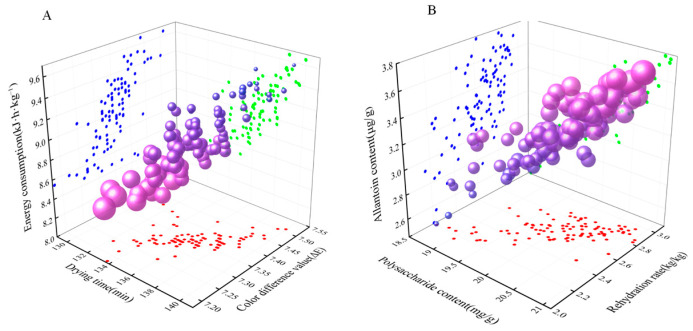
Pareto optimal set of optimal solutions. Note: (**A**) shows the relationship between drying time, unit energy consumption, color difference. (**B**) shows the relationship between rehydration ratio, polysaccharide content, allantoin content.

**Figure 21 foods-13-00434-f021:**
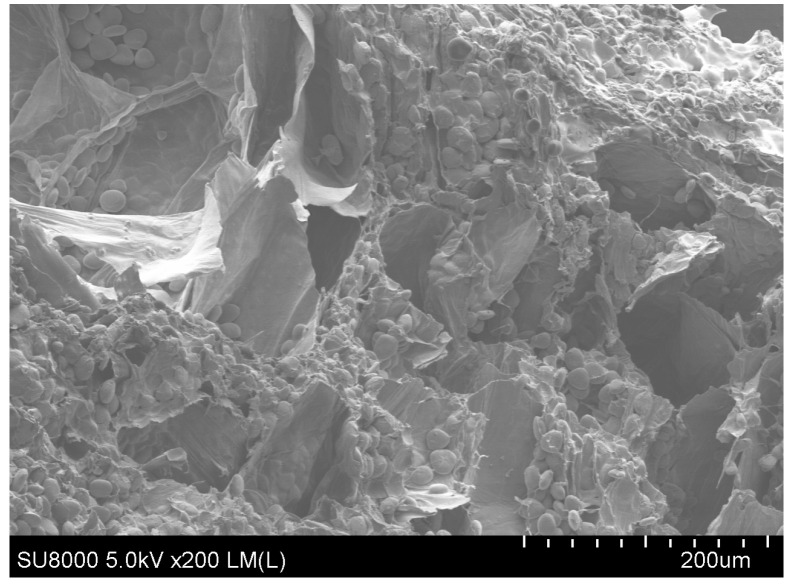
Microstructure of optimized yam slices.

**Figure 22 foods-13-00434-f022:**
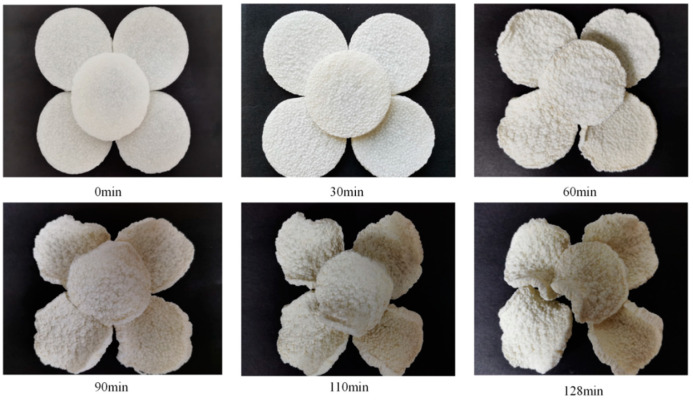
Drying process after optimization of yam slices.

**Table 1 foods-13-00434-t001:** Response surface factor level code table.

Code	Drying Conditions		
	Temperaturesa/°C	Slice Thicknessb/mm	Radiation Distancec/mm
−1	55	4	80
0	60	6	120
1	65	8	160

**Table 2 foods-13-00434-t002:** Linear regression equations and effective moisture diffusion coefficients of IR-HAD in yam slices at different temperatures.

Temperature/°C	Linear Equation	*R* ^2^	*D_eff_* (m^2^/s)
50	Ln*MR* = −0.2375*t* + 0.1338	0.99051	8.55 × 10^−9^
55	Ln*MR* = −0.2747*t* + 0.1072	0.99605	9.89 × 10^−8^
60	Ln*MR* = −0.3275*t* + 0.175	0.98857	1.18 × 10^−8^
65	Ln*MR* = −0.3673*t* + 0.1585	0.99189	1.32 × 10^−8^
70	Ln*MR* = −0.4097*t* + 0.1467	0.99437	1.79 × 10^−8^

**Table 3 foods-13-00434-t003:** Linear regression equations and effective water diffusion coefficients of IR-HAD for yam slices with different slice thicknesses.

Slice Thickness/mm	Linear Equation	*R* ^2^	*D_eff_* (m^2^/s)
2	Ln*MR* = −0.4978*t* + 0.1338	0.99276	1.79 × 10^−8^
4	Ln*MR* = −0.3984*t* + 0.1072	0.98842	1.43 × 10^−8^
6	Ln*MR* = −0.3275*t* + 0.175	0.98857	1.18 × 10^−8^
10	Ln*MR* = −0.1644*t* + 0.1467	0.98028	5.92 × 10^−9^

**Table 4 foods-13-00434-t004:** Linear regression equation and effective water diffusion coefficient of IR-HAD for yam slices at different radiation distances.

Radiation Distance/mm	Linear Equation	*R* ^2^	*D_eff_* (m^2^/s)
80	Ln*MR* = −0.43398*t* + 0.1338	0.97689	1.56 × 10^−8^
120	Ln*MR* = −0.3275*t* + 0.175	0.98857	1.18 × 10^−8^
160	Ln*MR* = −0.29663*t* + 0.1585	0.98885	1.07 × 10^−8^

**Table 5 foods-13-00434-t005:** Response surface analysis experimental design and results.

Code	A/°C	B/mm	C/mm	Y_1_ (min)	Y_2_ (ΔE)	Y_3_ (kJ·h/kg)	Y_4_ (mg/g)	Y_5_ (kg/kg)	Y_6_ (μg/g)
1	55	4	120	195	5.35	8.58	22.43	2.74	3
2	65	4	120	150	7.12	11.62	18.76	2.65	2.78
3	55	8	120	255	6.25	10.35	19.36	2.47	2.7
4	65	8	120	240	8.16	11.74	16.81	2.25	2.46
5	55	6	80	225	5.78	8.69	21.73	2.37	3.15
6	65	6	80	165	7.58	9.64	17.47	2.14	2.93
7	55	6	160	225	3.15	10.68	25.41	2.47	3.6
8	65	6	160	240	5.42	12.24	18.64	2.21	3.22
9	60	4	80	150	7.36	6.74	21.62	2.84	3.08
10	60	8	80	255	8.89	8.87	19.84	2.41	2.87
11	60	4	160	195	5.79	8.97	24.73	2.93	3.62
12	60	8	160	240	6.56	10.85	21.44	2.51	3.31
13	60	6	120	195	5.41	8.81	24.47	2.91	3.64
14	60	6	120	195	5.36	8.74	25.35	2.87	3.63
15	60	6	120	210	4.97	9.46	22.53	2.96	3.52
6	60	6	120	210	5.14	9.17	25.47	2.92	3.63
17	60	6	120	180	5.52	8.63	26.23	2.84	3.71

**Table 6 foods-13-00434-t006:** ANOVA of regression models and model terms.

Source	*df*	Y_1_ (min)		Y_2_ (ΔE)		Y_3_ (kJ·h/kg)		Y_4_ (mg/g)		Y_5_		Y_6_ (μg/g)	
		*F*-Value	*p*-Value	*F*-Value	*p*-Value	*F*-Value	*p*-Value	*F*-Value	*p*-Value	*F*-Value	*p*-Value	*F*-Value	*p*-Value
Model	9	14.68	0.0009	71.17	<0.0001	15.87	0.0007	9.74	0.0033	57.70	<0.0001	64.72	<0.0001
A	1	10.59	0.014	159.03	<0.0001	27.85	0.0012	23.77	0.0018	33.43	0.0007	34.14	0.0006
B	1	86.42	<0.0001	47.60	0.0002	20.11	0.0028	8.13	0.0246	120.69	<0.0001	39.43	0.0004
C	1	10.59	0.014	199.95	<0.0001	44.80	0.0003	7.30	0.0305	6.77	0.0353	88.21	<0.0001
AB	1	1.73	0.23	0.104	0.7567	3.15	0.1193	0.20	0.6679	1.77	0.2256	0.0060	0.9406
AC	1	10.80	0.0134	1.17	0.3153	0.430	0.5328	1.01	0.3491	0.0940	0.7680	1.53	0.2565
BC	1	6.91	0.0339	3.06	0.1238	0.071	0.7974	0.36	0.5652	0.0104	0.9215	0.5964	0.4652
*A* ^2^	1	2.01	0.1996	1.15	0.3192	45.77	0.0003	29.00	0.0010	245.80	<0.0001	196.00	<0.0001
*B* ^2^	1	0.55	0.4823	215.38	<0.0001	0.117	0.7424	12.88	0.0089	0.0027	0.9596	200.46	<0.0001
*C* ^2^	1	2.01	0.1996	8.93	0.0203	0.649	0.4471	1.38	0.2792	92.08	<0.0001	1.67	0.2376
Lack of Fit	3	0.59	0.6507	0.912	0.5106	2.92	0.1634	0.48	0.7142	1.26	0.3992	0.7793	0.5639
C.V. (%)	5.5			3.56		4.83		5.71		1.87		2.01	
*R* ^2^	0.9497			0.9892		0.9533		0.9260		0.9867		0.9881	

Note: *p* < 0.05 indicates significance; *p* < 0.001 indicates high significance.

**Table 7 foods-13-00434-t007:** Single-objective optimization results.

Evaluation Indicator	Optimization Direction	Factor			Optimal Value
		A/°C	B/mm	C/mm	
Drying times (min)	Minimum	63.41	4.58	81.04	136.37
Color difference (ΔE)	Minimum	55.00	5.80	160.00	3.30
Unit energy consumption (kJ·h·kg^−1^)	Minimum	58.16	4.00	80.00	6.75
Polysaccharide content (mg/g)	Maximum	57.68	5.18	159.69	26.23
Rehydration ratio	Maximum	58.35	5.15	117.96	2.97
Allantoin content (μg/g)	Maximum	59.42	5.98	147.93	3.81

**Table 8 foods-13-00434-t008:** Indicator weights.

Evaluation Indicator	Standard Deviation	Average Value	Coefficient of Variation	Weight
Drying times (min)	32.64	207.35	0.16	0.18
Color difference (ΔE)	1.34	6.11	0.22	0.25
Unit energy consumption (kJ·h·kg^−1^)	1.38	9.63	0.14	0.16
Polysaccharide content (mg/g)	2.95	21.9	0.13	0.15
Rehydration ratio	0.27	2.62	0.1	0.12
Allantoin content (μg/g)	0.38	3.23	0.12	0.13

**Table 9 foods-13-00434-t009:** Comparison of response surface and BO-GWO model results.

Mold	Y_1_		Y_2_		Y_3_		Y_4_		Y_5_		Y6	
	*R* ^2^	RMSE	*R* ^2^	RMSE	*R* ^2^	RMSE	*R* ^2^	RMSE	*R* ^2^	RMSE	*R* ^2^	RMSE
RSM	0.9667	5.9510	0.9794	0.1926	0.9568	0.2869	0.9502	0.6580	0.9766	0.0417	0.9754	0.0596
BP-GWO	0.9950	2.3005	0.9942	0.1020	0.9983	0.0573	0.9976	0.1445	0.9916	0.0250	0.9933	0.0312

**Table 10 foods-13-00434-t010:** Pareto10 set of optimal solution sets.

Serial Number	Drying Conditions			Evaluation Indicators					
	A/°C	B/mm	C/mm	Y_1_	Y_2_	Y_3_	Y_4_	Y_5_	Y_6_
1	64.51	4.26	90.95	133.68	7.18	8.54	20.79	2.78	3.64
2	64.21	4.34	92.49	133.41	7.22	8.57	21.08	2.83	3.71
3	62.08	4.21	90.44	133.90	7.26	8.46	20.61	2.68	3.45
4	63.55	4.32	91.87	132.23	7.24	8.54	20.72	2.96	3.77
5	64.06	4.31	92.25	133.80	7.27	8.52	21.04	2.84	3.67
6	63.39	4.20	90.36	134.12	7.30	8.51	20.81	2.82	3.77
7	62.94	4.30	91.92	134.99	7.30	8.53	20.80	2.87	3.75
8	64.14	4.28	91.33	133.18	7.27	8.64	20.73	2.89	3.75
9	62.58	4.21	90.02	133.80	7.27	8.52	20.65	2.84	3.78
10	64.21	4.29	92.26	133.99	7.29	8.56	20.10	2.87	3.65
Average value	63.57	4.27	91.39	133.71	7.26	8.54	20.73	2.84	3.69

**Table 11 foods-13-00434-t011:** Prediction and validation of response variables under optimal conditions.

	A/°C	B/mm	C/mm	Y_1_	Y_2_	Y_3_	Y_4_	Y_5_	Y_6_
Predicted values	63.57	4.27	91.39	133.71	7.26	8.54	20.73	2.84	3.69
Experimental values	64	4	91	128	7.54	8.61	19.94	2.76	3.63
Error (%)	-			4.46	3.71	0.81	3.96	2.9	1.65

## Data Availability

The original contributions presented in the study are included in the article, further inquiries can be directed to the corresponding author.

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
