# Peer review of "Quality and Process Optimization of Infrared Combined Hot Air Drying of Yam Slices Based on BP Neural Network and Gray Wolf Algorithm"

_foods, 2024, doi:10.3390/foods13030434_

Round 1
Reviewer 1 Report
Comments and Suggestions for Authors
The manuscript “Quality and process optimization of infrared combined 2 hot air drying of yam slices based on BP neural network and 3 gray wolf algorithm” is interesting and well written. However, the manuscript requires minor revision (i.e., the following points ) .
1. The manuscript requires English revision to eliminate ambiguity.
2. The authors should describe how they measure total energy consumption.
3. In the methodology section 2.14 GWO-BP model and multi-objective optimization, the author should describe the model parameters according to the experimental parameters.
4. Line 333 conFig.d?
5. Figure 21 translates the axis title in English.
6. In the conclusion section author should draw concluding remarks on the key findings.
Comments on the Quality of English LanguageThe manuscript requires English revision to eliminate ambiguity.
Reviewer 2 Report
Comments and Suggestions for Authors
Comments on the Quality of English LanguageModerate editing of English language required
Author Response
Please refer to the annex

Round 2
Reviewer 2 Report
Comments and Suggestions for Authors
The authors responded clearly to all the comments raised therefore the manuscript can be accepted.
Author Response
Thanks to the reviewers for their support, we will definitely continue to work hard!